# Potential Refinement of Recurrence Score by pSTAT3 Status

**DOI:** 10.3390/genes13030438

**Published:** 2022-02-27

**Authors:** Albert Grinshpun, Yogev Cohen, Aviad Zick, Luna Kadouri, Tamar Hamburger, Benjamin Nisman, Tanir M. Allweis, Gabriela Oprea, Tamar Peretz, Beatrice Uziely, Amir Sonnenblick

**Affiliations:** 1Sharett Institute of Oncology, Hadassah-Hebrew University Medical Center, Jerusalem 91120, Israel; albert.grinshpun@mail.huji.ac.il (A.G.); yogev.cohen@mail.huji.ac.il (Y.C.); aviadz@hadassah.org.il (A.Z.); luna@hadassah.org.il (L.K.); tamarh@hadassah.org.il (T.H.); nisman@hadassah.org.il (B.N.); tamary@hadassah.org.il (T.P.); beatrice@hadassah.org.il (B.U.); 2Faculty of Medicine, The Hebrew University, Jerusalem 9112102, Israel; tanir@hadassah.org.il; 3Department of Surgery, Hadassah-Hebrew University Medical Center, Jerusalem 91120, Israel; 4Department of pathology, Emory University, Atlanta, GA 30322, USA; goprea@emory.edu; 5Institute of Oncology, Tel Aviv Sourasky Medical Center, Tel Aviv, Israel and Sackler Faculty of Medicine, Tel Aviv University, Tel Aviv 6423906, Israel

**Keywords:** early breast cancer, recurrence score, oncotype, pSTAT3, prognosis

## Abstract

The likelihood of recurrence in breast cancer patients with hormone receptor-positive (HR-positive) tumors is influenced by clinical, histopathological, and molecular features. Recent studies suggested that activated STAT3 (pSTAT3) might serve as a biomarker of outcome in breast cancer patients. In the present work, we have analyzed the added value of pSTAT3 to OncotypeDx Recurrence Score (RS) in patient prognostication. We have found that patients with low RS (<26) and low pSTAT3 might represent a population at a higher risk for cancer recurrence. Furthermore, we have observed that a positive pSTAT3 score alone can be a favorable marker for patients with HR-positive breast cancer under the age of 50. In an era of personalized medicine, these findings warrant further appraisal of chemotherapy benefit in this population.

## 1. Introduction 

Recent advances in the field of personalized medicine have deepened our understanding of how to integrate clinical parameters with molecular tools to prognosticate patients. When looking into clinical management of early breast cancer, routine use of molecular assay such as OncotypeDX Breast Recurrence Score (RS) has revolutionized clinical practice and helps clinicians to assess the risk of relapse and the need, if necessary, for systemic adjuvant therapies [1,2].

Yet, molecular tools are not perfect, and their accuracy is a matter of debate [3]. Therefore, multiple additional tools are being studied to aid the prognostication process and provide clinically relevant information of the benefit of adjuvant chemotherapy [4,5,6]. 

Janus Kinases (JAKs) and their downstream signal transducers and activators of transcription (STAT) proteins are heavily involved in malignant processes. Specifically, STAT3 was found to be crucial for tumor progression through different mechanisms that include direct effects on proliferation, angiogenesis, apoptosis, and survival, as well as having a non-direct effect on the immune and stromal environment surrounding the tumor [7,8,9,10,11].

In patients with breast cancer, STAT3 has several key roles in tumor development and progression, as shown by numerous studies [4,7,8,12,13,14]. Previously, we and others have shown the prognostic and predictive roles of phospho-STAT3 (pSTAT3) in breast cancer, mainly in luminal tumors [4,5,12,13,14,15,16].

In the current preliminary work, we sought to determine whether pSTAT3 can add additional prognostic information to RS results and guide better clinical decision making. 

## 2. Methods

### 2.1. Tissue Analysis:

Formalin fixed, paraffin-embedded invasive breast cancer tissues from hormone receptor (HR) positive, HER2-negative, early breast carcinoma patients treated between the years 1996–2013 were available from the Department of Pathology.

Tissue cores with a diameter of 0.6 mm were taken with a manual tissue array maker MTA-1 (Beecher Instruments) arrayed in duplicates or triplicates on a recipient paraffin block and placed on charged poly-lysine-coated slides [12].

The immunodetection and scoring of pSTAT3 was carried out as previously described, and the staining was determined according to the the Remmele score (0–6, intensity + percentage, intensity (0, negative; 1, weak; 2, moderate; 3, strong) + percentage of the stained tumor cells (0, <10%; 1, 10–25%; 2, 25–50% and 3, 50–100%)) [12].

Nuclear Phospho-STAT3 staining was determined separately for each specimen (Figure 1).

Positive pSTAT3 IHC was defined as score ≥ 2.

### 2.2. Clinical Data

Demographic, clinical, and recurrence data were obtained from electronic medical records. The use of tumor specimens and data in this research was approved by the Medical Ethics Committee of the Hadassah Medical Center, and each patient gave written informed consent.

### 2.3. Statistical Analysis

The statistical analysis was generated using SPSS software version 27. *Chi-square* test for categorical data and unpaired Student’s t-test for continuous variables were used. Estimated DFS and overall survival (OS) rates according to the pSTAT3 score subgroups were calculated by the product limit method (Kaplan–Meier) for all patients and were stratified by age at diagnosis; 95% confidence intervals (CIs) were calculated from the model. Multivariate models were computed with Cox proportional-hazards regression.

*p* values ≤ 0.05 were considered statistically significant. 

## 3. Results 

The study population comprised of 449 women with interpretable pSTAT3 IHC. The median age at diagnosis was 52. A comparison between pSTAT3 positive versus pSTAT3 negative tumors did not demonstrate a significant relationship between pSTAT3 positivity and age, histological grade, tumor size, histological type, and involved lymph node number (*p* > 0.05). Furthermore, we have analyzed the relation between pSTAT3 status and survival in 436 patients with known outcome. The median follow up was 105 months (censored at 10 years of follow up). In pSTAT3-positive tumors, the 10-year overall survival was non-significantly greater than pSTAT3 negative tumors (78.1% vs. 69.4%, *p* = 0.17). A multivariate analysis adjusted for stage, age, and grade did not demonstrate an association of pSTAT3 with overall survival (hazard ratio for death, 0.72 (95% CI, 0.41–1.28); *p* = 0.26).

To obtain additional insight into the role of pSTAT3 in prognostication, we have analyzed a subset of 75 patients with known RS results (Table 1). Of them, 16 patients (21.3 %) had breast cancer recurrence, and one patient suffered from metastatic second primary tumor (lung cancer). The median age on primary diagnosis was 70.15. The median follow-up time was 114 months.

Nine patients (12%) had a high RS of 26 or more. Additionally, 36% of patients had positive pSTAT3 scores (≥2). It is noteworthy that patients in low and high RS groups (≤25, 26 and more, respectively) had similar characteristics in regard with age, tumor grade, histological type, tumor size, and lymph node number. 

When patients were divided according to their pSTAT3 IHC scores (Figure 2a), we observed that positive scores were significantly associated with a reduced risk of recurrence (*p* = 0.005); out of the 16 patients that had disease recurrence, only one was positive for pSTAT3. 

Additionally, patients’ allocation with negative pSTAT3 scores (<2) to high or low RS (with a cutoff of 26) showed that both groups have similar percentage of recurrence.

On the other hand, when patient allocation started according to RS scores (Figure 2b) and then we dichotomized according to positive or negative pSTAT3, the low RS\high pSTAT3 group had a significantly better prognosis (*p* = 0.007).

DFS analysis according to the pSTAT3 score did not show a statistically significant difference between the ‘negative’ groups and the ‘positive’ group (Figure 3a, *p* = 0.109). Yet, DFS analysis according to the pSTAT3 score showed a statistically significant difference between the ‘negative’ groups and the ‘positive’ group when adjusted by age at diagnosis (Figure 3b, *p* = 0.03).

Multivariate analysis of the clinical and pathological data for DFS was preformed (Table 2). We found that several clinical and pathological characteristics were associated with worse DFS, including age under 50 at diagnosis (*p* = 0.039) and Lobular type (*p =* 0.024).

The hazard ratio of RS score (<26/>26) is 0.548 (*p* = 0.518), and the hazard ratio of pSTAT3 score is 5.462 (*p* = 0.137).

## 4. Discussion 

In our preliminary analysis, we have partially recapitulated the known association between positive pSTAT3 to better prognosis and found that pSTAT3 scores might potentially add valuable prognostic data to RS in assessing patients’ risk of early, HR-positive breast cancer recurrence. 

As STAT3 functions as a key player in numerous pro-tumorigenic processes, one may expect that its activation will be associated with poor prognosis. However, the reality is far more complex, as shown in previous publications on pSTAT3 as being both a poor and improved prognosis biomarker. Interestingly, this phenomenon was also shown in other tumor types, such as STAT3 contradictory roles in non-small lung cancer [17].

STAT3, as was mentioned, has been shown to play key roles in the cell’s cycle and is mostly considered growth promoting and an antiapoptotic factor. In normal conditions, STAT3 activation is terminated by suppressors of cytokine signaling proteins. 

However, cancer cells exhibit constitutively activated STAT3, which is attributable to upregulated tyrosine kinases and the deregulation of its negative regulation by suppressors of cytokine signaling proteins. 

STAT3 activation and its target genes in the cell stimulate growth, angiogenesis, and cell motility and inhibit apoptosis. In addition to cell survival and proliferation, STATs regulate several other biological processes that may contribute to cancer. STAT3 has been shown to promote angiogenesis and may contribute to cancer by allowing tumors to evade detection by the host immune system

One of the important cytokines in this mechanism is interleukin-6 (IL6). 

IL6, binding to its receptor complex IL6R/gp130, activates downstream Janus kinases, which subsequently activate the signal transducer and activator of STAT3 through phosphorylation.

Interestingly, IL6/STAT3 signaling has been shown to play a role in tumor progression by inducing epithelial to mesenchymal transition and angiogenesis. In breast cancer, STAT3 is most often activated by IL6 [18].

In fact, many breast cancer cell lines produce IL6, which activates STAT3 by signaling through the IL-6 receptor and Jak kinases [18,19].

One would expect that tumors that express constitutive phospho-STAT3 expression would be associated with poorer prognosis, but our results show otherwise.

In order to resolve this seemingly troublesome inconsistency, we theorized that in clinic, pSTAT3 has a predictive role for adjuvant chemotherapy benefit in breast cancer patients.

Dien et al. found a trend, although the results were not statistically significant for longer survival in patients with increased STAT3 activation, and hypothesized that pSTAT3 regulates Tissue Inhibitor of Metalloproteinase-1 (TIMP1), which makes breast cancer cells less invasive [20]. 

Furthermore, other studies showed that STAT3 is a tumor suppressor protein with a role in breast tissue cellular differentiation and apoptosis [21,22].

Couto et al. found that siRNA knockdown of STAT3 resulted in significantly increased tumor growth in thyroid cancer cell lines. This study also found a correlation between the STAT3 knockdown and increased glucose uptake and the production of lactate in the cell lines. Those finding suggest that one of the mechanisms of the tumorigenesis inhibition character of STAT3 is mediated by inhibiting aerobic glycolysis [23].

Finally, Lee et al. found that STAT3 mediates tumor suppressor effects by binding to GSK3β, which, in turn, promotes the phosphorylation and the degradation of Snail, a critical regulator of the EMT and cancer metastasis [24].

Evidence that STAT3 plays a role in cellular differentiation and apoptosis, functions as a tumor suppressor, and regulates the in-cell process that decreases cancer cell invasiveness may be consistent with better outcomes in breast cancer patients with high pSTAT3.

A possible explanation for our findings can be drawn from a recent computational analysis of pSTAT3 phenotype [5]. In that work, researchers have shown that patients in the luminal A population were much more likely to possess a pSTAT3 high phenotype, whereas those in the luminal B population were much more likely to have a pSTAT3 low phenotype. 

## 5. Conclusions

In modern oncology, clinicians’ decisions on adjuvant chemotherapy are complex and based on a variety of factors: clinical, pathologic, and molecular. Yet, much room for improvement and personalization exists. In the present trial, we correlate pSTAT3 to RS and identify another and novel potential population (low RS\low pSTAT3) that might benefit from chemotherapy despite low RS (<26). Furthermore, pSTAT3 immunohistochemistry as a simple stand-alone test may possibly also provide prognostic data [4,5,25,26,27].

Taken together, our observations imply that a double-low score (low RS\low pSTAT3) is a marker of unfavorable outcome in HR-positive breast cancer, mainly in patients under the age of 50 on primary diagnosis. If confirmed in a larger-designed prospective randomized trial, our findings would indicate that pSTAT3 testing could be used as an adjunct biomarker to direct rational adjuvant treatment in luminal breast cancer patients.

## Figures and Tables

**Figure 1 genes-13-00438-f001:**
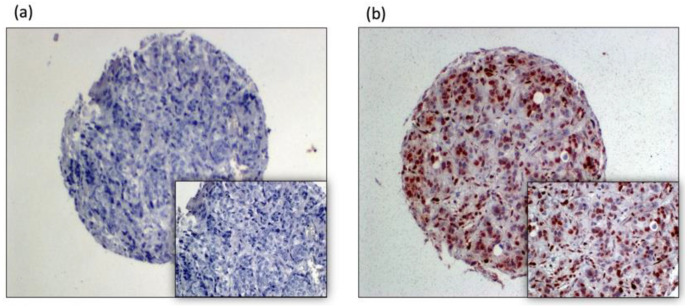
pSTAT3 expression, breast cancer tissue microarray sports as follows: (**a**) pSTAT3 negative, (**b**) pSTAT3 strongly positive.

**Figure 2 genes-13-00438-f002:**
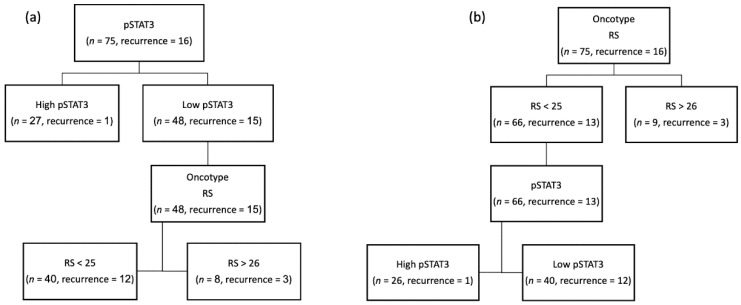
(**a**) Patients divided according to their pSTAT3 IHC scores; (**b**) patient allocation started according to RS scores.

**Figure 3 genes-13-00438-f003:**
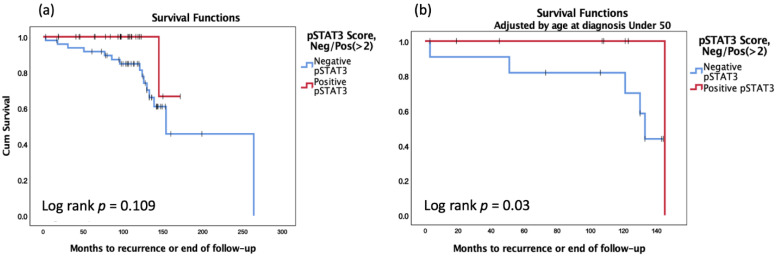
Kaplan–Meier analysis for DFS (**a**) according to the pSTAT3; (**b**) according to the pSTAT3 and adjusted by age at diagnosis.

**Table 1 genes-13-00438-t001:** Patient characteristics according to pSTAT3 Score.

		pSTAT3 Score, Negative (<2)/Positive (≥2)				
Variable		*n* = 75				
Sub-Variable		Negative		Positive		*p* Value
		(*n* = 49)	n%	(*n* = 26)	n%	
Age at diagnosis	Over 50	38	77.6%	18	69.2%	0.43
	Under 50	11	22.4%	8	30.8%	
Tumor type	Ductal	35	71.4%	17	65.4%	0.966
	Lobular	5	10.2%	3	11.5%	
	Mucinous	1	2.0%	1	3.8%	
	Both Ductal and Lobular	8	16.3%	5	19.2%	
Lymph Nodes	No	38	77.6%	20	76.9%	0.951
	Yes	11	22.4%	6	23.1%	
Chemotherapy	No	33	67.3%	20	76.9%	0.357
	Yes	16	32.7%	6	23.1%	
Hormone therapy	No	6	12.2%	2	7.7%	0.706
	Yes	43	87.8%	24	92.3%	
RS score	≤26	41	83.7%	25	96.2%	0.15
	>26	8	10.3%	1	3.8%	
Recurrence	No	34	69.4%	25	96.2%	0.007
	Yes	15	30.6%	1	3.8%	
Age	Median (range)	61.39 (29.7–81)		57.91 (38.6–77.9)		0.963
	Mean ± SD	58.01 ± 12.07		57.88 ± 10.68		

**Table 2 genes-13-00438-t002:** Multivariate analysis for DFS.

Multivariate Analysis of the Clinical and Pathological Data for DFS				
	Adjusted HR	95.0% CI for HR		*p* Value
		Lower	Upper	
RS score, Neg/Pos (>26)	0.548	0.089	3.394	0.518
Lymph Nodes	2.709	0.405	18.119	0.304
Type: Ductal/Lobular/Mucinous/Ductal & Lobular				0.024
Type: Ductal vs. Lobular	14.367	1.261	163.673	0.032
Type: Ductal vs. Mucinous	39.831	2.747	577.569	0.007
Type: Ductal vs. Ductal and Lobular	3.839	0.893	16.502	0.071
Chemotherapy	0.556	0.118	2.623	0.458
Hormone therapy	1.946	0.302	12.532	0.483
Tumor Size (cm)	0.981	0.438	2.198	0.963
pSTAT3 Score, Negative/Positive (≥2)	5.462	0.584	51.121	0.137
Age at diagnosis under 50	5.778	1.092	30.560	0.039

## Data Availability

The data presented in this study are available on request from the corresponding author.

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
