# Peer review of "Potential Refinement of Recurrence Score by pSTAT3 Status"

_genes, 2022, doi:10.3390/genes13030438_

Round 1
Reviewer 1 Report
The authors presented an original work on the prognostic role of STAT3 IHC expression in patients with ER+ and HER2-negative breast cancer. At a first analysis, STAT3 did not showed a discriminative prognostic role and was not associated with known prognostic features. However, a subgroup analysis showed an interaction with age and in patients with low-oncotypeDX risk score. The findings are consistent with previous publications, and serve as a confirmation study.
The authors could speculate on the mechanisms of resistance, including the enhancers of the STAT3 response. For example, a role of IL-6 has been suggested.
If available, pictures of pSTAT3 at IHC might be helpful.
Author Response
Dear Reviewer,
Thank you so much for your insights.
Please see the attachment.

Reviewer 2 Report
The manuscript "Potential Refinement of Recurrence Score by pSTAT3 Status" is an interesting study in the scientific and clinical aspects.
After a brief introduction, the Authors placed the chapter on Methods. While the Material chapter may be replaced by Table 1 and Figure 1a, the methods should be described in detail, despite the archival material used for the research. Due to the significant importance of STAT3 gene expression as a clinical marker, I believe that the manuscript "Potential Refinement of Recurrence Score by pSTAT3 Status" should be accepted for publication in GENES after supplementing the methods used.
Author Response

(The authors gave the same response as above.)
